# Pathology of Combined Hepatocellular Carcinoma-Cholangiocarcinoma: An Update

**DOI:** 10.3390/cancers15020494

**Published:** 2023-01-13

**Authors:** Florian Roßner, Bruno Valentin Sinn, David Horst

**Affiliations:** 1Institute of Pathology, Charité—Universitätsmedizin Berlin, 10117 Berlin, Germany; 2German Cancer Consortium (DKTK), Partner Site Berlin and German Cancer Research Center (DKFZ), 69120 Heidelberg, Germany

**Keywords:** combined hepatocellular and cholangiocellular carcinoma, hepatocellular carcinoma, cholangiocarcinoma, mixed carcinoma, cHCC-CCA, stem cells, progenitor cells, hippo signaling, wnt signaling

## Abstract

**Simple Summary:**

Combined hepatocellular carcinoma-cholangiocarcinoma (cHCC-CCA) is a rare tumor with its origin in the liver, risk factors comparable to hepatocellular carcinoma and poor prognosis. This review describes its histomorphological classification and diagnosis over time and outlines the molecular foundation for potential prospective therapeutic approaches.

**Abstract:**

Combined hepatocellular carcinoma-cholangiocarcinoma (cHCC-CCA) is a rare primary liver cancer that is composed of both hepatocellular and cholangiocellular differentiated cells. It is slightly more common in men and among Asian and Pacific islanders. Overall, risk factors are similar to classic risk factors of hepatocellular carcinoma (HCC). The classification has significantly evolved over time. The last WHO classification (2019) mainly emphasized diagnosis on morphological basis with routine stainings, discarded previously recognized classifications with carcinomas with stem cell features, introduced intermediate cell carcinoma as a specific subtype and considered cholangiolocarcinoma as a subtype of cholangiocellular carcinoma. Immunohistochemical markers may be applied for further specification but have limited value for diagnosis. Recent discoveries in molecular pathway regulation may pioneer new therapeutic approaches for this poor prognostic and challenging diagnosis.

## 1. Introduction

Primary liver cancers are the sixth-most common malignancies (4.7%, both sexes) and remarkably the third-most common cause of cancer mortality (8.3%) after colorectal and lung cancer worldwide [1]. Incidence and age of onset vary substantially by geography: Asia (72.5%) has the highest incidence followed by Europe (9.7%), Africa (7.8%), North America (5.1%), Latin America and the Caribbean (4.4%), and Oceania (0.5%) [2]. Among primary liver malignancies, hepatocellular carcinoma (HCC) is the most common type [3]. Mean ages at diagnosis are generally above the sixth decade in North America and Europe, whereas in Asia and Africa, mean ages range between the third and sixth decades [3]. Common risk factors are liver cirrhosis, viral hepatitis B and viral hepatitis C infection, metabolic conditions like nonalcoholic fatty liver disease (NAFLD) [3,4] and environmental exposure to aflatoxins [5], thorotrast [6], anabolic steroids [7], oral contraceptives [8,9,10], alcohol [11], or tobacco [12,13,14]. Furthermore, developmental diseases like Abernethy malformation [15] and other conditions based on gene mutations and loss of function, e.g., Alagille syndrome [16], ataxia teleangiectasia [16,17], bile salt export protein deficiency [16,18], and tyrosinemia [16] bear a higher risk for development of hepatocellular carcinoma. A small fraction of hepatocellular adenomas may progress into hepatocellular carcinoma [19].

Cholangiocellular carcinoma (CCC) is the second-most common primary liver cancer, accounting for 15% of primary liver malignancies [20]. Hepatolithiasis and recurrent pyogenic cholangitis are strong risk factors for the development of cholangiocarcinoma [21,22]. Some risk factors are shared with HCC. Viral hepatitis B and viral hepatitis C infections [23,24,25], alcohol consumption, and tobacco smoking [23,26] are known risk factors, although affiliations may not be as strong as in cases with hepatocellular carcinoma [20]. CCC is most common in East Asia due to parasite infections [27]. Furthermore, cholangiocellular carcinoma-specific predisposing factors like autoimmune primary sclerosing cholangitis (PSC) [28,29,30,31] and congenital anomalies of the biliary tree like Caroli’s syndrome, congenital hepatic fibrosis, and choledochal cysts carry an approximate 15% risk of malignant transformation into CCC in adulthood [32,33].

Besides pure HCC and CCC, combined hepatocellular and cholangiocellular carcinoma (cHCC-CCA) represents a rare, yet highly interesting entity of primary liver cancer, which tends to occur in middle-aged men with a slight predominance in people with an Asian Pacific background. Due to its rareness, significance of statements on risk factors are generally limited in literature but factors similar to HCC like male gender and viral hepatitis B infections are observed. On a histomorphological basis, the tumor presents mixed features of hepatocellular and cholangiocellular carcinoma. Recent morphological and molecular pathology studies suggest a common cell of origin and opened novel insights into subordinate molecular pathway structures [34,35,36]. However, there is still a distinct lack of knowledge in this field. This review summarizes important cornerstones of several decades of investigation, the evolution of perspectives on histological classification, recent advances of molecular pathologic analysis and gives a brief outlook on potential future approaches. 

## 2. Materials and Methods

For this narrative review, we conducted a comprehensive literature search using both general and specific terms covering different research areas. General terms were “combined”, “etiology”, “prognosis”, “immunohistochemistry”, “subtypes”, “expression”, “genesis”, and “prognostic”. Specific terms for etiological issues were “cirrhosis”, “viral hepatitis”, “infectious”, “toxic”, “metabolic”, “genetic”, “environmental”, “demography”, “ethnic”, and “ethnicity”. Specific terms for immunohistochemical issues were “ihc”, “immunostains”, “immunohistochemistry”, “immunohistochemical”, “expression”, “expression marker” and “biomarker”. Specific terms for molecular issues were “small interfering rna”, “sirna”, “rnai”, “epigenetics”, “molecular”, “driver mutation”, “mutation”, “fusion”, “rearrangement”, “amplification”, “signaling”, “pathway”, “carcinogenesis”, “alteration”, “deficiency”, “deletion”, “insertion”, “phosphorylation”, “acetylation”, “next generation sequencing”, “ngs”, “plasmid”, “hras”, “myc”, “p53”, “YAP”, “Hippo”, “Notch”, “Wnt”, “PIK3K”, “Notch”, “Nfkb”, “Jak”, “Stat”, “liquid biopsy”, and “circulating tumor cells”. Synonyms for “combined hepatocellular and cholangiocellular carcinoma” comprised also “cholangiocellular hepatocellular carcinoma”, “hepatocellular-cholangiocarcinoma” and “hepatocellular-cholangiocellular carcinoma”. These terms were integrated into our search matrix. All search terms were also combined and applied to contents in titles and abstracts. Overall, we obtained 2450 results, whereof 913 results were double hits. Further refinement and elimination of broadly unrelated literature left 729 citations of which 242 are cited in this review (Figure 1). Original articles were obtained where possible. The literature search was performed with Endnote X9. Figures were created with Adobe Illustrator 2022. 

## 3. Results

### 3.1. Epidemiology

The incidence of combined hepatocellular and cholangiocellular carcinoma (cHCC-CCA) among primary liver cancer lesions is reported to range from 0.4–14.2% [37,38,39,40,41]. One of the largest series of primary liver lesions comprised 465 patients diagnosed with cHCC-CCA and found an incidence of 0.77% [37]. Another large study comprised 529 patients with cHCC-CCA and reported an overall incidence of 0.05 per 100,000. Overall, men had a higher incidence than women (0.08 vs. 0.03 per 100,000 per year). These tumors were more common among Asian and Pacific islanders (0.08 per 100,000) than among Whites and Blacks (0.05 per 100,000) and American Indian/Alaska Natives (0.04 per 100,000). Remarkably, the incidence seemed to rise during the 10-year study period [41]. The incidence of cHCC-CCA among different ethnic groups is reported in the literature with contrasting results. Wachtel et al. conducted a study of a 30-year-period SEER database set with 282 patients diagnosed with cHCC-CCA who showed a higher incidence of hepatomas than cHCC-CCA in patients with Asian/Pacific origin. Men were less likely than women to develop cholangiocarcinomas than cHCC-CCA. Men, more than women, and patients of Asian/Pacific ethnicity, more than patients born elsewhere, were more likely to develop hepatomas than cHCC-CCA [42]. Sex predilection for the male gender was already described in early studies with smaller cohorts [38,43,44]. However, all known figures on incidences seem to bear an uncertain inherent bias because most patients did not undergo surgical resection and thus may have been misdiagnosed with HCC and CCC [37]. Taken together, cHCC-CCA is a rare entity with higher incidence in males with Asian and Pacific background. However, surveys may show conflicting results due to its rare nature.

### 3.2. Risk Factors and Prognostic Factors

Statements on risk factors in cHCC-CCA are generally limited due to its rare nature and inconsistent diagnostic classification for several years. Risk factors for cHCC-CCA are comparable to those of hepatocellular carcinoma (HCC), arguing that characteristics of clinicopathological features rendered cHCC-CCA rather a variant of HCC than being a true intermediate entity between HCC and CCC. cHCC-CCA patients showed similarities with HCC regarding male predilection, status of viral hepatitis, serum alpha-protein (AFP) level and nontumor histology [45,46,47]. Furthermore, primary biliary cholangitis (PBC), a well-known risk factor for HCC, also seemed to be a rare precondition for the development of cHCC-CCA [24,48]. However, these results turned out to be controversially discussed. Demographic and clinical features were reported to be more similar to CCC in some smaller cohorts [39,49]. 

Prognostic influences in cHCC-CCA included sex, tumor size, and therapy status with worse prognosis in patients who did not receive surgical resection, radiation therapy, or chemotherapy [41,50,51]. Advanced disease with nodal and distant spread also compromised survival likelihoods [37,46,50,51]. Garancini et al. described a predilection of African Americans with worse disease-specific survival odds [37].

### 3.3. Basic Clinical Aspects

Reported clinical features mainly relied on retrospective studies with small cohorts and little statistical power [52]. In general, cHCC-CCA is usually silent until it presents with advanced disease and symptoms that are also typical for HCC or CCC, including painless jaundice, fatigue, abdominal discomfort, weight loss, pruritus, ascites, acute cholangitis, fever, and hepatomegaly [53]. Combined cHCC-CCA may have distinct imaging characteristics, dependent on the predominant differentiation but interpretation may be challenging [52]. Staging of cHCC-CCA is adapted to the staging of cholangiocarcinomas. Cholangiocarcinomas may be classified after the Bismuth–Corlette classification, which comprises perihilar cholangiocarcinomas [54]. More up-to-date classification proposals are in fact published with the combined American Joint Committee on Cancer (AJCC)/Union for International Cancer Control (UICC) cancer staging manual, which comprises classification systems for distal bile duct tumors [55], perihilar tumors [56], and intrahepatic tumors [57]. However, there has been criticism of the Bismuth–Corlette classification concerning information on vascular encasement and distant metastases and about the TNM staging system that vessel invasion affects T stage so that stage groupings do not serve as a guide for choosing local therapy. As a result, a new staging system for perihilar tumors was proposed, which encompasses the size of the tumor, the extent of the disease in the biliary system, the involvement of the hepatic artery and portal vein, the involvement of lymph nodes, distant metastases, and the volume of the putative remnant liver after resection [58]. 

Advanced tumor stages spread by metastases in lymph nodes and several parenchymal organs, whereas comprehensive data may be missing due to the rareness of the entity. Maeda et al. describe seven autopsy cases with metastases in the lungs, adrenal gland, spleen, peritoneum, skin, bone, brain, and lymph nodes of porta hepatis, hilum of the lung, supraclavicle, paratrachea, perigastric, parapancreas, and para-aorta [44]. Twenty autopsy cases by Goodman et al. showed metastasis most often in the lung and portal lymph nodes [59]. 

Currently, minor or major hepatic resection, with or without lymph node dissection, is the only curative option and the most studied approach cHCC-CCA [37,46,47,60,61,62,63,64,65,66,67,68,69,70,71,72,73,74], whereas outcomes of a liver transplantation show controversial results compared to resection in data currently available [37,60,75]. Comparative data on locoregional treatment options like transarterial chemoembolization (TACE), which are considered in inoperable cases is limited to a few retrospective studies [76,77,78] and benefits remain debatable. Patients with advanced disease should be treated according to the principles established for advanced cholangiocarcinoma [79] and usually receive a gemcitabine and platin-based regimen in first-line treatment [79,80,81,82,83].

The prognosis is generally thought to be intermediate between that of pure HCC and intrahepatic cholangiocarcinoma, but tend to be closer to cholangiocellular carcinoma [37,46,53,84,85]. Lee et al. found overall survival rates at one, three, and five years of 55%, 15%, and 5% after resection, respectively. The corresponding values for CCC were 48.1%, 18.5%, and 0% and 67%, 40%, and 21% for HCC [47]. 

### 3.4. Histopathological Features

#### 3.4.1. A Historical Perspective

Combined hepatocellular and cholangiocellular carcinoma displays features of both hepatocytic and cholangiocytic differentiation. These were already well described in the early 20th Century and known under various terms: mixed tumors [86,87], intermediate type [88,89], carcinoma of dual origin [90], cholangiohepatoma [91], and duplex type [92]. The first case was described by Wells in 1903 [93]. Although histomorphological characteristics between the two principal forms of liver cancer had been described by Winternitz [94] and others, the immediate anatomical neighbourhood of bile ducts, hepatocytes, and fluent morphological transitions in cells with features intermediate between hepatocytic and cholangiocytic phenotypes caused confusion in recognition [95] and classification [96,97,98] for several years. Allen and Lisa published a small case series in 1949 [99] and concluded that three histomorphological combinations of tumors of double nature were possible: (1) separate neoplastic masses, (2) contiguous tumors, each of which has a different cell type, which may mingle as they grow, and (3) individual lesions that have both types of cells so intimately associated that they can only be interpreted as arising from the same site. In 1954, a small number of these types of cancers were discussed by Edmondson and Steiner in a series of 100 cases of primary liver cancer, given a separate place and name (hepatobiliary carcinoma), but were numbered among hepatocellular carcinomas due to their proposed hepatocellular origin [96]. In 1959, Steiner et al. proposed—although admittedly a rare incidence—a tumor with cholangiolocellular phenotype and coined the term “cholangiolocellular carcinoma” for this type of carcinoma, which is thought to be derived from epithelium of the cholangioles or canals of Hering [100]. Goodman et al. proposed a new classification into the three groups of “collision tumors” (Type I), which reflects an apparent coincidental occurrence of both hepatocellular carcinoma and cholangiocarcinoma in the same patient, "transitional tumors” (Type II), in which there are areas of intermediate differentiation and an identifiable transition between hepatocellular carcinoma and cholangiocarcinoma and “fibrolamellar tumors” (Type III), which resembled the fibrolamellar variant of hepatocellular carcinoma but bears mucin-producing pseudoglands [59].

#### 3.4.2. The WHO Classification

The original World Health Organization (WHO) classification [101] considered mucin production to be evidence of biliary differentiation and so a tumor with unequivocal hepatocellular differentiation and either gland formation by biliary type epithelial cells or a positive mucin stain was considered to be a hepatocellular-cholangiocellular carcinoma. This category should not be used for those tumors in which either form of growth is insufficient for identification.

The fourth edition of the WHO classification of Tumours of the Digestive System again defined unequivocal, intimately mixed elements of both hepatocellular carcinoma and cholangiocellular carcinoma as combined hepatocellular-cholangiocarcinoma in contrast to HCC and CCC arising in the same liver, which may also be intermixed similar to Goodman’s collision tumors [59]. Over the course of time, several studies suggested combined hepatocellular and cholangiocellular tumors to be accompanied by tumor cells with stem cell features [36,102,103,104,105,106,107], so for the first time the WHO recognized different subtypes with stem cell features. The classical type remains the most typical form of cHCC-CCA and is defined by a well-, moderate or poorly differentiated hepatocellular component and a good, moderate, or poor biliary component, often accompanied with abundant stroma (Figure 2). In many of these mixed hepatobiliary carcinomas, there are foci of intermediate morphology at the interface of HCC and CCC components (Figure 3 and Figure 4). Either component may further be highlighted by appropriate immunohistochemical stains and features of stemness may also be demonstrable by immunohistochemistry. Furthermore, there are three subtypes with stem-cell features defined, although they have not been considered distinct entities at this time because it has since then become uncertain whether they bear biological differences at all. cHCC-CCA with stem-cell features of the classical type features nests of mature appearing hepatocytes accompanied by clusters of small cells with high nuclear to cytoplasmatic ratio and hyperchromatic nuclei [36,103]. These nests may be separated by fibrous stroma to impart a scirrhous appearance. cHCC-CCA with stem cell features of intermediate subtype features tumor cells with intermediate features between hepatocytes and cholangiocytes [104]. These tumor cells are small with an oval-shaped hyperchromatic nucleus and grow primarily with a solid-trabecular pattern in a desmoplastic stromal background. Moreover, ill-defined elongated gland-like structures may be present. cHCC-CCA with stem cell features of cholangiolocellular type marks the opposite end of the histomorphologic spectrum (Figure 5). These small cells form populations in tubular, cord-like, anastomosing patterns, also called “antler-like” patterns, and also feature high nuclear-to-cytoplasmic ratios and hyperchromatic, oval nuclei [105,108]. They are reported to recapitulate cholangioles or canals of Hering. This subtype has been traditionally categorized under cholangiocellular carcinomas but has been classified as combined tumor in the fourth edition of the WHO classification. This has changed again in the fifth edition. 

The fifth edition of the WHO classification simplifies the categories (Table 1). Essential features of a cHCC-CCA are biphenotypic primary liver cancers which show unequivocal features of both hepatocytic and cholangiocytic differentiation based on H&E morphology, whereas immunophenotypic expression patterns are not sufficient by themselves for diagnosis but can be used as an additional method to confirm cell differentiation. In a consensus paper, Brunt et al. justifiably argued that so-called cancer stem cells could be found in various proportions in the transitional zone between the two components, which express a wide variety of immunohistochemical markers. These markers are not entirely specific for a distinct differentiation so that classifications of tumors with stem cell features are no longer recommended [109]. Nevertheless, it is recognized that some primary liver cancers show monotonous cells with features intermediate between hepatocellular and cholangiocellular carcinoma and therefore keep the designation as so-called intermediate cell carcinoma [104,109,110]. For the diagnosis of intermediate cell carcinoma, the tumor has essentially to consist of unequivocal intermediate cells in the entire tumor [110]. Cholangiolocellcarcinoma can be present within a cHCC-CCA; however, a sole cholangiolocellcarcinoma or an admixture with a conventional cholangiocarcinoma is now considered a component of cholangiocarcinoma, which is also supported by molecularpathological features [35].

### 3.5. Immunohistochemical Landscape

#### 3.5.1. General Considerations

Confirmation of cell differentiation is usually provided by immunohistochemical staining of differentiation markers. Hepatocellular differentiation is more often displayed by granular cytoplasmatic expression of Hep Par1, cytoplasmatic expression of arginase 1 [111], canalicular staining of CD10 and polyclonal expression of carcinoembryonic antigen (CEA) [112,113,114,115,116,117]. Alpha-fetoprotein may also be expressed but is reported to lack sensitivity similar to EMA, B72.3, and others [113,118] and has even been investigated in canine primary liver cancers [119]. To highlight biliary structures, cytokeratin 7 [44] and cytokeratin 19 [120] may be useful markers. However, they do not confirm biliary differentiation because hepatocellular components may also be positive [121,122]. Most importantly, no hepatocellular or biliary differentiation markers are 100% specific. Moreover, positivity of classic differentiation markers for allegedly hepatocellular differentiation has also been described in cholangiocellular carcinoma [123,124,125]. In fact, the vast majority of differentiation markers are expressed in hepatocellular, cholangiocellular, and intermediate cells to different degrees (see Table 2), so that in the first place a prudent analysis of the histomorphological features on H&E stain is advisable together with an array of multiple immunohistochemical stains to obtain a proper diagnosis. 

#### 3.5.2. Immunohistochemical Expression Patterns of Conventional Differentiation Markers

Evidence of hepatocyte paraffin 1 (Hep Par 1) is one of the most used markers for hepatocellular differentiation and is described in both hepatocellular and cholangiocellular elements of combined hepatocellular and cholangiocellular carcinoma [126]. Hep Par 1 was expressed predominantly in hepatocellular differentiated areas of cHCC-CCC (53.1% [127], 52.9% [128]), but was also demonstrable in the cholangiocellular differentiated area in one study [128]. Nevertheless, a relatively high sensitivity and specificity for hepatocellular differentiation is attributed to Hep Par 1 [129]. Hep Par 1 can be used in a diagnostic panel together with arginase-1 and cytokeratin 7, 8, 18, and 19 when it comes to differentiating hepatocellular elements, intermediate cell types, and cholangiocellular elements on a immunohistochemical basis [130].

Alpha-fetoprotein (AFP) may be used to confirm hepatocellular origin. Several studies showed low or moderate positive staining rates in hepatocellular tumor components (24% [118], 29% [59], 46.2% [127]), but it lacks sensitivity and may also be positive in cholangiocellular differentiated cells [126].

Akiba et al. described differentially expressed Arginase-1 in a small series of HCC, CCC and cHCC-CCA. Arginase-1 was expressed in all HCC cases (100%) but only in one CCC (20%). cHCC-CCA showed Arginase-1 expression only in the hepatocellular component (80%, classical subtype; 69%, intermediate subtype, 64% cholangiolocellular subtype), whereas it can also be found in cholangiocellular cells (20%). However, expression in mixed tumor components was significantly lower than in HCC [130].

Carcinoembryonic antigen (CEA) may be a useful marker to stain bile canaliculi in otherwise hepatocellular differentiated carcinomas and supports evidence for cholangiocellular differentiation [131]. Brumm et al. described CEA expression in cholangiocellular carcinoma and cholangiocellular differentiated areas of cHCC-CCA [132]. Ganjei et al. described exclusive expression in CCC [133]; however, conflicting reports have also been published. Tickoo et al. described a canalicular pattern in hepatoid areas throughout, and Goodman et al. reported mixed expression in both components of cHCC-CCA [59]. 

Cytokeratin AE1/AE3 is broad spectrum pancytokeratin marker for Type I and Type II keratins [134,135] with substantive overlaps in detection patterns of cytokeratin 5.2 [136]. Both antibodies detect hepatocellular parenchyma and bililary epithelium, and were described to extensively stain both components of cHCC-CCA [113,118,126].

Cytokeratin 7 (CK7) and cytokeratin 19 (CK19) rank among the most used antibodies to differentiate ductal epithelial elements like bile ducts. Indeed, several authors describe high rates of CK7 expression in CCC, but CK7 is also described in HCC [126,137,138]. Maeda et al. found CK7 expression in 100% of CCC and 50% of HCC. CK7 was simultaneously expressed in 100% of tumor cells of cholangiocellular differentiated component of cHCC-CCC and in 24% of hepatocellular component [44]. Other authors found 100%, 80%, 58.3% in CCC component and 100%, 52%, and 25% in hepatocellular component, respectively [118,127,137]. Other authors did not give differentiated statements about expression profiles but stated 52.9% in both hepatocellular and biliary differentiated zones [128], 80% and 100% in cHCC-CCA [138,139]. Kim et al. reported CK7 in one out of 12 cases (8.3%) [140]. Cytokeratin 7 is also described in transitional zones of so-called intermediate cells [141]. A similar picture can be drawn for cytokeratin 19 staining. Cytokeratin 19 is predominantly expressed in biliary type epithelium with rates of 85% to 100% in CCC, whereas a minority of HCC stained for CK19 (0–30% respectively) [44,114,138,140,142]. In combined tumors, the cholangiocellular differentiated areas were positive in 83–100%, whereas staining in hepatocellular areas was highly variable in 8.3–100% [44,114,127,138,139,140]. Pozharisskii et al. noted a sensitivity for CCC of 83% and a specificity of 78% [129].

Cytokeratin 8 (CK8) expression is accompanied by a high prevalence in cholangiocellular differentiated areas, similar to cytokeratins mentioned above (80–100%) [126,130]. Akiba et al. additionally described a common staining of so-called intermediate cells and cholangiocellular areas in cHCC-CCA (78.1% and 82%) [130]. In difference, cytokeratin 18 (CK18) was unequivocably expressed in both biliary and hepatocellular differentiated tissues [126,130].

Cytokeratin 20 (CK20) plays a subordinate role in the differential diagnosis of HCC and CCC. Staining is exhibited in a minority of cases (6.2% HCC, 9.4% CCC) [114] and also stains a minority of cHCC-CCA. CK20 may be useful in differential diagnosis of questionable mestastic disease [114]. 

To summarize, despite a wide array of studied biomarkers, in a cost effective and yet sensitive approach, Hep Par 1 and arginase-1 seem to be most useful for displaying hepatocyte differentiation, whereas CK7 and CK19 may be used to depict cholangiocellular differentiation.

#### 3.5.3. Immunhistochemical Expression Patterns of Stem Cell and Progenitor Cell Markers

Overall, epithelial cell adhesion molecule (EpCAM) seems to be preferentially expressed in biliary type epithelium. Variable expression between 18.2% and 75% has been reported in cHCC-CCA in general [138,140], 14.3% and 88.5% in hepatocellular differentiated components and 71.4% and 100% in cholangiocellular components specifically [127,143]. Interestingly, cells of so-called cHCC-CCA with stem cell features of intermediate type frequently express EpCAM [143,144], whereas cells in typical subtype and cholangiocellular subtype cancers only show focal staining [144]. 

CD133 tended to be more highly expressed in cholangiocellular components of cHCC-CCA and in the intermediate cell type than in HCC areas (14.3% vs. 42.2%) and together with EpCAM expression profile has been shown to be associated with poor prognosis [143]. Ikeda et al. similarly found significant expression rates in intermediate-type cells (41.7%), whereas expression in typical and cholangiocellular subtypes were variable. CD133, together with other markers like CD56 and DLK1 had lower expression rates in intermediate type cells, which may reflect a state of transdifferentiation rather than cell stemness [144].

Delta-like noncanonical notch ligand 1 (DLK1) expression was found in stem-cell-like cells of cHCC-CCC (66.7% in typical subtype, 41.7% in intermediate subtype and 83.3% in cholangiocellular subtype). Expression in HCC and HCC-like areas was highly variable between 10.9% and 56.8% [144,145,146]. It remained unclear whether DLK1-positivity means stem cell/progenitor origin, because putative oval cells did not express DLK1 in nonlesional liver. However, the discussion proceeds as features typical for stem cells likely increased chemoresistance, colony formation, spheroid colony formation, and in vivo tumorigenicity, and were attributed to DLK1-positive cells [147].

Epithelial membrane antigen (EMA) overall was more commonly expressed in glandular elements of cHCC-CCA and cholangiocarcinoma than in hepatocellular elements [118,126,127,138], but the difference in expression did not reach significant levels [118].

Tyrosine-protein kinase KIT (c-KIT) is widely accepted to suggest a stem cell or progenitor phenotype [148]. However, c-KIT expression was shown to be expressed in all compartments of c-HCC-CCA in a case report, strongly suggesting that this tumor was entirely derived from liver stem cells/progenitor cells [126]. This is in line with other reports. High c-Kit expression is generally described in transitional, intermediate areas of c-HCC-CCA [127,138,139]. Other authors report, in addition to high expression rates in intermediate type cells [104], expression in hepatocellular carcinoma [104] and cholangiocellular carcinoma [138].

In addition to c-Kit, neural cell adhesion molecule (NCAM/CD56) is a popular marker described to reflect stem cell/progenitor cell features, whereas similar to c-Kit, high expression rates in hepatocellular-like areas [127], cholangiocellular-differentiated areas [138] and transitional areas with intermediate type cells [141,144] are reported. 

Nestin has recently been found to be highly expressed in intermediate cells of cHCC-CCA, cholangiocellular-differentiated areas of cHCC-CCA as well as cholangiocarcinomas [141,149].

CD44 issued limited staining results in one study (26%), but may be considered as a prognostic marker [140]. Cytokeratin 34ßE12 is expressed in cholangiocarcinomas with basal cell type features [150].

#### 3.5.4. Other Investigated Markers 

In addition to the already-mentioned marker array, several other biomarkers have been investigated over time, which remained so far in an experimental approach and did not reach broad acceptance, likely due to a lack of specificity. 

Alpha-1-antitrypsin and fibrinogen expression was described by Goodman et al. in the mid1980s and was highly expressed in hepatocellular carcinoma, cholangiocellular carcinoma, and cHCC-CCA [59]. Alpha-human chorionic gonadotropin showed limited expression in HCC and no expression in cHCC-CCA [132]. Carboanhydrase IX (CA IX) has been tested in HCC and CCC, but showed mixed results [151]. Patil et al. showed high expression results for C-reactive protein (CRP) in both hepatocellular and cholangiocellular component of cHCC-CCA but staining lacked sensitivity and specificity [152]. Fibrin stabilizing factor (Factor XIIIa) was shown to be variably expressed [153] and in one study failed to show expression at all [114]. For all we know, metallothionein was not examined in cHCC-CCA, but showed variable results in HCC, and correlated with tumor progression [154,155,156]. Mucins were extensively examined by Sasaki et al. and others. MUC1 showed high expression in cholangiocarcinomas and cholangiocellular differentiated areas of cHCC-CCC [157,158,159], whereas MUC2 [158,159], MUC3 [158,159], MUC5/6 [159], MUC5AC [158], MUC6, and MUC7 [158] showed limited usefulness for further differentiation. Hepatocellular progenitor cells were also identified by applying p63, polyductin and prominin-1 [105,150,160]. Vimentin was expressed in HCC with sarcomatoid features [115,132,161].

YAP-1 and TAZ are key effector proteins in the hippo signaling pathway and therefore interesting markers for investigation in cHCC-CCA. Van Haele et al. published a comprehensive study, in which cytokeratin 19-positive HCC showed significantly higher nuclear YAP-1 and TAZ than cytokeratin-negative HCC cases. All cases of cHCC-CCA and CCC showed nuclear YAP-1 and TAZ. HCC with YAP and TAZ expression and cHCC-CCA were also linked to poor disease-free survival (DFS) and poor overall survival (OS) [162]. 

A summary of antigens demonstrable in tumor cells of c-HCC-CCA is given in Table 2, whereas the most commonly used and most specific antigens for hepatocellular and cholangiocellular differentiation are listed separately. In contrast, no specific stem cell surrogate marker seems to exist, and a wide variety of biomarkers is studied as of today.

**Table 2 cancers-15-00494-t002:** Summarization of some antigens demonstrable in tumor cells of cHCC-CCA ^1^.

Tumor	Immunohistochemical Stains	References
Most commonly used biomarkers		
Hepatocellular cells	Hepatocyte Paraffin 1 (Hep Par 1)	[104,114,126,127,128,129,137,138,139,153,163]
	Arginase 1	[111,130]
Cholangiocellular cells	Cytokeratin 7	[44,118,126,127,128,137,138,139,140,141,163]
	Cytokeratin 8	[126,164,165]
	Cytokeratin 19	[44,104,114,118,127,129,138,139,140,141,142,163,166]
Further studied biomarkers		
Hepatocellular cells	Albumin	[164]
	Alpha-1-antichymotrypsin	[164]
	Alpha-1-antitrypsin	[59,164]
	alpha-Fetoprotein	[59,104,113,118,126,127,132,133,138,144,163]
	Alpha-human chorionic gonadotropin	[132,164]
	Asialoglycoproteinreceptor	[164]
	Biliary glycoprotein	[164]
	ß-microglobulin	[164]
	CA IX	[151]
	Carcinoembryonic antigen (CEA) polyclonal	[59,104,113,118,126,131,132,133]
	Cathepsin B	[164,167,168,169,170]
	CD44	[140]
	C-reactive protein (CRP)	[152,164]
	Cytokeratin 18	[126,164,165]
	Cytokeratin 20	[114,118,126]
	Cytokeratin AE1/AE3	[113,118,126]
	Cytokeratin CAM 5.2	[113,118,126]
	Delta-like 1 homolog (DLK1)	[144]
	Epithelial membrane antigen (EMA)	[113,118,126,127,138]
	EpCAM	[126,127,138,140,143,144,171]
	Erythropoiesis-associated antigen (ERY-1)	[133,164]
	Factor XIIIa	[114,153,164,172]
	Ferritin	[164]
	Fibrinogen	[59,164]
	Metallothionein	[154,155,156,164]
	P21	[164]
	P53	[126,164]
	Proliferating cell nuclear antigen (PCNA)	[164,171]
	Prothymosin	[164,173]
	Retinoblastoma	[129,164]
	Thrombospondin	[164]
	Transferrin receptor	[164]
	Ubiquitin	[164]
	Vimentin	[115,132,161,164]
	Yes-associated protein 1 (YAP1)	[162,174,175,176]
Intermediate cells	Alpha-Fetoprotein (AFP)	[59,104,113,118,126,127,132,133,138,144,163]
	Arginase 1 (ARG1)	[111,130]
	Carcinoembryonic antigen (CEA) polyclonal	[59,104,113,118,126,131,132,133]
	CD133	[140,143,144]
	CK34ßE12	[126,150]
	c-KIT	[104,105,138,139,150]
	Cytokeratin 7	[44,118,126,127,128,137,138,139,140,141,163]
	Cytokeratin 8	[126,164,165]
	Cytokeratin 18	[126,164,165]
	Cytokeratin 19	[44,104,114,118,127,129,138,139,140,141,142,163,166]
	Delta-like 1 homolog (DLK1)	[144]
	EpCAM	[126,127,138,140,143,144,171]
	Hepatocyte Paraffin 1 (HepPar 1)	[104,114,126,127,128,129,137,138,139,153,163]
	LIF	[105]
	Nestin	[141,149,177]
	Neural cell adhesion molecule (NCAM)/CD56	[126,127,138,141,143,144]
	OCT4	[105]
	P63	[150]
	Polyductin	[160]
	Prominin-1	[105]
	Yes-associated protein 1 (YAP1)	[162,174,175,176]
Cholangiocellular cells	ARID1A	[178]
	CA IX	[151]
	Carcinoembryonic antigen (CEA) polyclonal	[59,104,113,118,126,131,132,133]
	c-KIT	[104,105,138,139,150]
	Cytokeratin 7	[44,118,126,127,128,137,138,139,140,141,163]
	Cytokeratin 19	[44,104,114,118,127,129,138,139,140,141,142,163,166]
	Cytokeratin AE1/AE3	[113,118,126]
	Cytokeratin CAM 5.2	[113,118,126]
	Delta-like 1 homolog (DLK1)	[144]
	Epithelial membrane antigen (EMA)	[113,118,126,127,138]
	EpCAM	[126,127,138,140,143,144,171]
	MUC1	[126,137,157,159,179]
	MUC2	[126,159,179]
	MUC3	[159,179]
	MUC5/6	[159]
	MUC5AC	[126,158]
	MUC6	[126,158]
	MUC7	[158]
	Neural cell adhesion molecule (NCAM)/CD56	[126,127,138,141,143,144]
	Polyductin	[160]
	Tumor-associated glyocoprotein 72 (TAG-72)	[180]
	Yes-associated protein 1 (YAP1)	[162,174,175,176]

^1^ Modified after [164].

### 3.6. Ultrastructural Features

Electron microscopy may show a biphasic differentiation of cuboidal to columnar neoplastic cells lying in close proximity to one another. Hepatocellular differentiation is defined by abundant organelles comprising mitochondria, rough endoplasmatic reticulum, lipid droplets, and glycogen particles, as well as accompanying bile canaliculi. Cholangiolar differentiated cells have a smooth nuclear outline combined with comparatively few organelles, a variable number of tonofilaments, microvillus projections, and junctional complexes between cells [181].

### 3.7. Molecular Pathological Foundations

Recent studies discovered several significantly mutated genes with different prevalence in hepatocellular carcinoma, intrahepatic cholangiocarcinoma and combined hepatocellular and cholangiocarcinoma (Figure 6). 

In one study, exome analysis of 243 HCC mutational signatures associated with specific risk factors. A total of 161 putative driver genes in 11 recurrent pathways, namely TERT promoter mutations activating telomerase expression (60%), WNT/ß-catenin (54%), PI3K/AKT/mTOR (51%), TP53/cell cycle (49%), MAP kinase (43%), hepatic differentiation (34%), epigenetic regulation (32%), chromatin remodeling (28%), oxidative stress (12%), Il6/JAK/STAT (9%), and TGFß (5%), aggregated in three gene groups centered around CTNNB1, TP53, and AXIN1 were found. TERT promoter mutation was identified as an early event in hepatocellular carcinogenesis, whereas FGF3, FGF4, FGF19/CCND1 amplification, TP53, and CDKN2A alterations appeared in advanced stages of more aggressive tumors [182]. 

ICC-specific molecular alterations were disclosed by Zou et al. with 25 significantly mutated genes and eight potential driver genes (TP53, KRAS, IDH1, PTEN, ARID1A, EPPK1, ECE2, and FYN) affecting three important pathways (KRAS/PI3K; p53-cell cycle signaling and TGFß/SMAD-signaling) in ICC. A total of 9713 synonymous and nonsynonymous mutations were found, averaging 94.3 mutations per patient, whereas the numbers of somatic mutations varied dramatically ranging from 16 to 1333 [183]. Genetic and epigenetic alterations are likely to show an etiology-dependent pattern. Integrative analysis of expression data suggests two biological classes of ICC. Inflammation class is characterized by activation of inflammatory signaling pathways, overexpression of cytokines, and STAT3 activation, whereas proliferation class is characterized by activation of oncogenic signaling pathways such as RAS, MAPK, and MET [184].

Inflammation-related alterations have been described as primary sclerosing cholangitis-associated cholangiocarcinoma. In one study, COX-2 was significantly upregulated in contrast to nonneoplastic bile duct epithelial cells and sporadic ICC [185]. Hepatitis B infection may be linked to p53-mutation [186]. Mutational patterns may also be site-specific. IDH1/2, BRAF mutations and FGFR2 fusions were found in small duct ICC and are linked to progenitor cell expansion and development of premalignant biliary lesions [187,188]. Molecular alterations in cHCC-CCA and identification of driver mutations which promote the progression of cHCC-CCA have long been neglected and remained unknown, potentially due to its rare occurrence. The molecular pathological landscape shows significant heterogeneity and conflicting results [34,35]. The molecular signature of stemness in cHCC-CCA led to a support of a conceptual stem/progenitor cell as common cellular origin in this mixed tumor [36,189,190]. Coulouarn et al. showed that cHCC-CCA exhibits stem/progenitor features on a molecular level, a downregulation of the hepatocyte differentiation program and a commitment to the biliary lineage mainly driven by TGFbeta- and Wnt/beta-catenin pathways. This signature was associated with microenvironment remodeling. These findings were significantly distinct from well-differentiated HCC and showed characteristics of poorly differentiated HCC with stem cell traits and poor prognosis [34]. Stem cell-like signatures were confirmed in a genomic analysis which showed spalt-like transcription factor 4 (SALL4) positivity, enrichment of progenitor-like signatures, activation of specific oncogenic pathways (i.e., MYC and insulin-like growth factor), whereas the cholangiolocellular-differentiated areas showed a distinct biliary-derived entity associated with chromosomal stability and active TGF-beta signaling [35]. 

In an early publication, the allelic status of chromosome arms was studied for loss of heterogeneity (LOH), so that either a collision tumor with two independent clones, a single clonal tumor with divergent differentiation potential, or a single clonal process with genetic heterogeneity as a result of clonal evolution—and accordingly paralleled histological diversity—is postulated [191]. Another study found high LOH sites in CCC and cHCC-CCA areas, specifically at 3p and 14q. Moreover, TP53 and CTNNB1 expression was compared between the distinct differentiated areas. It was concluded that cHCC-CCA and CCC are genetically closer than cHCC-CCA and HCC [192], which is in line with previous findings. Nonetheless, the molecular landscape of LOH remains highly fragmentated. Combined hepatocellular-cholangiocarcinoma shared frequent +1q (71%), +8q (57%), and −8p (57%) with hepatocellular carcinoma, but a tendency for higher numbers of imbalance with cholangiocarcinoma. Cholangiocarcinoma was characterized by combined losses at 6q and 3p and a tendency for chromosomal instability. Combined hepatocellular-cholangiocarcinoma may share similar chromosomal changes with both hepatocellular carcinoma and cholangiocarcinoma, as reflected by common hepatocellular carcinoma-like +8q, +1q, and −8p and a tendency for cholangiocarcinoma-like chromosomal instability [193]. 

Further evidence of current heterogeneity in the molecular landscape is provided by a study, which in contrast showed a mutational pattern closer to HCC than CCC. Joseph et al. found a high percentage of TERT alterations (80%), TP53 mutations (80%), alterations in cell cycle genes (40%; CCND1, CCNE1, CDKN2A), receptor tyrosine kinase/Ras/PI3-kinase pathway genes (55%; MET, ERBB2, KRAS, PTEN), chromatin regulators (20%; ARID1A, ARID2), and Wnt pathway genes (20%; CTNNB1, AXIN, APC), even in the CCC component. TERT alterations were detected in both components, suggesting an early event in CHC evolution. ICC demonstrated alterations in IDH1 or IDH2 (30%), CDKN2A (40%), FGFR2 (20%), PBRM1 (20%), ARID1A (10%), and BAP1 (10%). These results suggested molecular profiles similar to HCC [194]. TP53 and TERT were the most frequently mutated genes (31.3% and 17.6% of samples, respectively). Mutations were also found in IDH1, IDH2, PIK3CA, and NRAS genes. Intermediate and HCC areas of cHCC-CCA seemed to share the same mutational profile, and both harbored different mutations than the CC component [141].

Whereas the molecular fingerprint of cHCC-CCA is largely shown to be associated with a more aggressive tumor type, environmental influences might be capable of shifting the genetic mutational landscape. Chronic hepatitis showed a strong impact on the mutational landscape in liver cancer and the genetic diversity among biliary differentiated liver cancers, where recurrent mutations in TERT promoter, chromatin regulators (BAP1, PBRM1 and ARID2), a synapse organization gene (PCLO), IDH genes and KRAS were identified. The frequencies of KRAS and IDHs mutations, which are associated with poor disease-free survival, were significantly higher in hepatitis-negative cases [195].

## 4. Discussion

In this review, we describe the winding diagnostic path of a rare primary liver lesion. Despite great progress in acknowledgement of this highly aggressive tumor over the last few decades, combined hepatocellular-cholangiocarcinoma urgently needs further investigational efforts to dissect its underlying nature. A first important step was the establishment of standards in diagnosis [109]. Furthermore, diagnosis should be rendered by experts in the field due to pitfalls in differential diagnosis especially on biopsy specimen [196].

Immunohistochemical markers currently in use lack sensitivity and specificity for a clear-cut diagnosis. This tumor group combines morphological and immunophenotypical hepatocellular and cholangiocellular characteristics with mixed characteristics in features of stemness within one tumor. In consequence, a recent consensus of international experts recommended a diagnosis based on routine histopathology with hematoxylin and eosin (H&E). Immunostains can be supportive but are not essential for diagnosis [109]. Well-established immunohistochemical markers like hepatocyte paraffin 1 and arginase-1 may be used to confirm hepatocytic differentiation, whereas Type I and Type II cytokeratins 7 and 19 may be used to confirm ductular or biliary differentiation, respectively [130]. In addition to common markers of lineage differentiation, a wide variety of markers, which are supposed to indicate cell stemness, were applied with variable specific results. Nestin has been shown to be expressed solely in stem cell-like intermediate type cells, and expression therefore should be considered when rendering a differential diagnosis [141].

In addition to specific diagnostic features, comprehensive knowledge about key regulatory molecular systematics in subcellular pathways will be critical for further effective therapeutic approaches. 

## 5. Future Directions

In recent years, several efforts to trace diagnosis and to disclose predictive and prognostic factors of hepatocellular carcinoma in a noninvasive manner have been developed. Liquid biopsy describes a technique to detect circulating tumor cells (CTC) by either indirect immunolabeling or reverse transcriptase-polymerase chain reaction (RT-PCR)-based methods [197]. Combined approaches with serum markers and measurement of CTC show promising results. Two studies significantly improved detection sensitivity in relatively large cohorts with 222 HCC patients and 395 HCC patients [198,199]. Although several studies proved high sensitivity and specificity values [138,198,200,201,202,203,204] and outperformed sole serum marker detection, sensitivity and specificity varies widely [205,206,207] and seems to be dependent on CTC characterization and choice of expression markers. In addition to CTC detection, great effort is invested in the detection of hepatocellular carcinoma with cell-free DNA (cfDNA) focused on the total amount of DNA [208,209,210,211,212,213,214,215], certain mutational patterns including TERT, TP53, CTNNB1, AXIN1, and other genes [216,217,218,219] and epigenetic methylation patterns [220,221,222,223,224,225,226,227,228,229,230,231,232,233,234,235,236,237,238,239,240]. Although cHCC-CCA has been shown to hold distinct molecular alterations [177], to the best of our knowledge, as of today, the benefits of liquid biopsy analysis have still to be proven in a respective study. 

Another promising noninvasive approach affects great advances in imaging analysis like convolutional neural networks [241] and automated imaging analysis. Schmauch et al. presented a deep learning algorithm to infer transcriptomic profiles from histological images, including genes involved in immune cell signaling and cancer-specific pathways, certainly capable of capturing even subtle textural changes [242]. Promissory evolutions like deep learning algorithms may be main drivers of future classifications and may help to pave the way for new therapeutic approaches. 

## 6. Conclusions

cHCC-CCA is a rare primary liver tumor and remains a challenging diagnosis despite advances in defining criteria and classification. Although great progress was made in molecular characterization, further studies will be crucial to decipher new and modern therapeutic approaches.

## Figures and Tables

**Figure 1 cancers-15-00494-f001:**
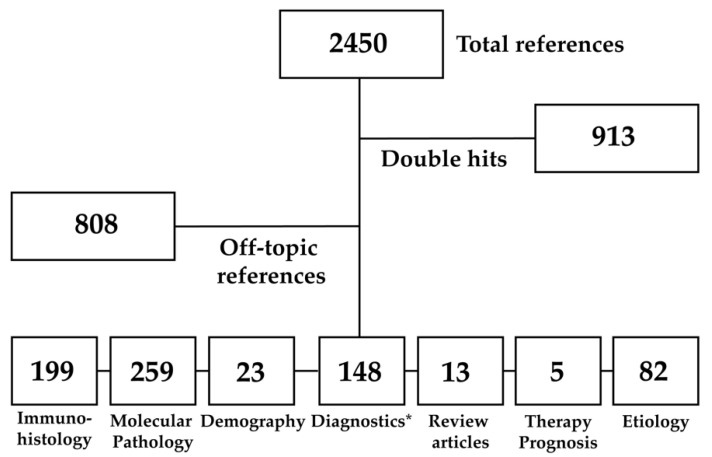
Summary of our literature search. * Diagnostics include noninvasive diagnostic approaches (laboratory diagnostics, medical imaging).

**Figure 2 cancers-15-00494-f002:**
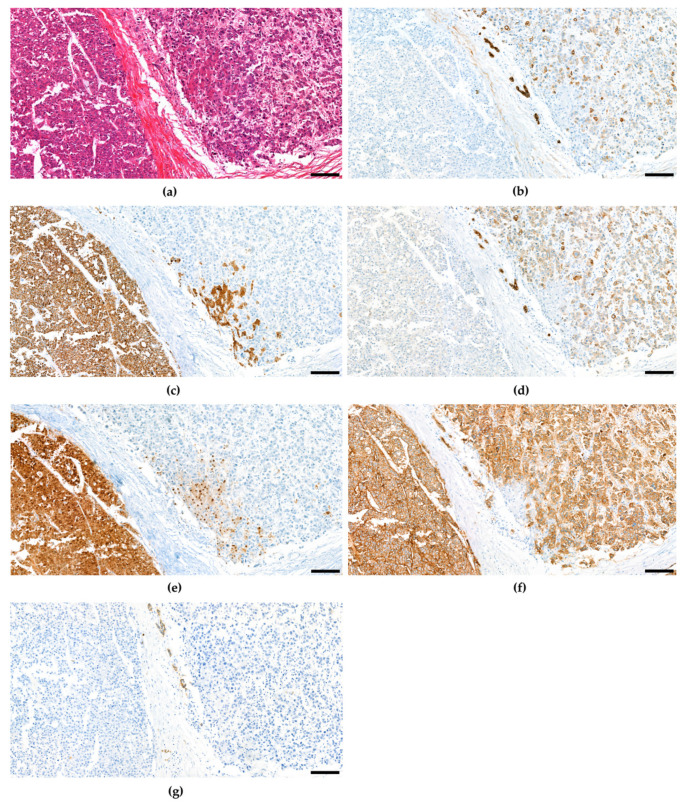
Combined hepatocellular and cholangiocellular carcinoma with distinct hepatocellular (left) and cholangiocellular (right) differentiated tumor components. (**a**) H&E. (**b**) CK7. (**c**) HepPar1. (**d**) CK19. (**e**) Arginase-1. (**f**) EpCAM. (**g**) CD56. Scale bar: 100 µm.

**Figure 3 cancers-15-00494-f003:**
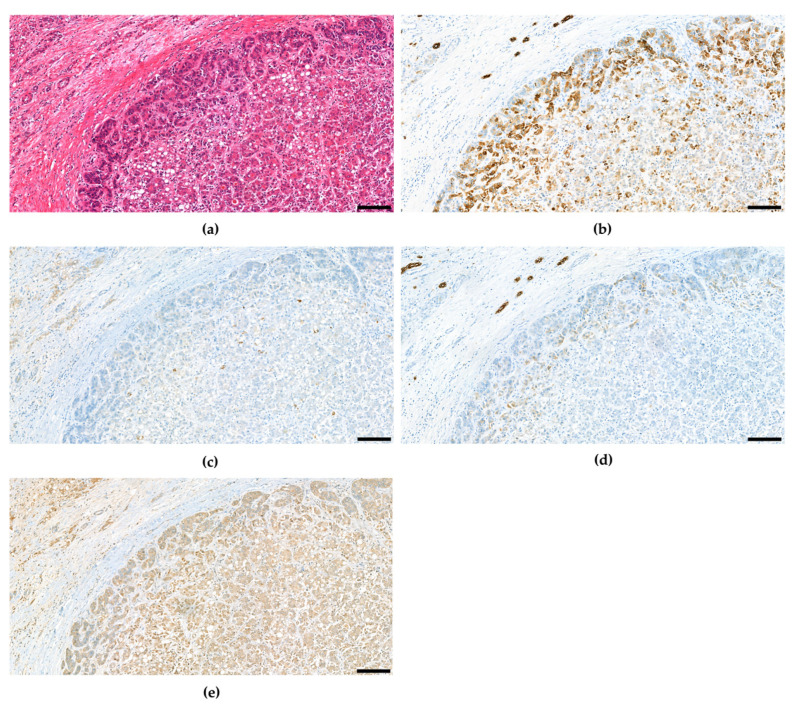
Combined hepatocellular and cholangiocellular carcinoma, transitional zone elements and intermediate cell features. CD56 and c-Kit were not expressed (not shown). (**a**) H&E. (**b**) CK7. (**c**) HepPar1. (**d**) CK19. (**e**) Arginase-1. Scale bar: 100 µm.

**Figure 4 cancers-15-00494-f004:**
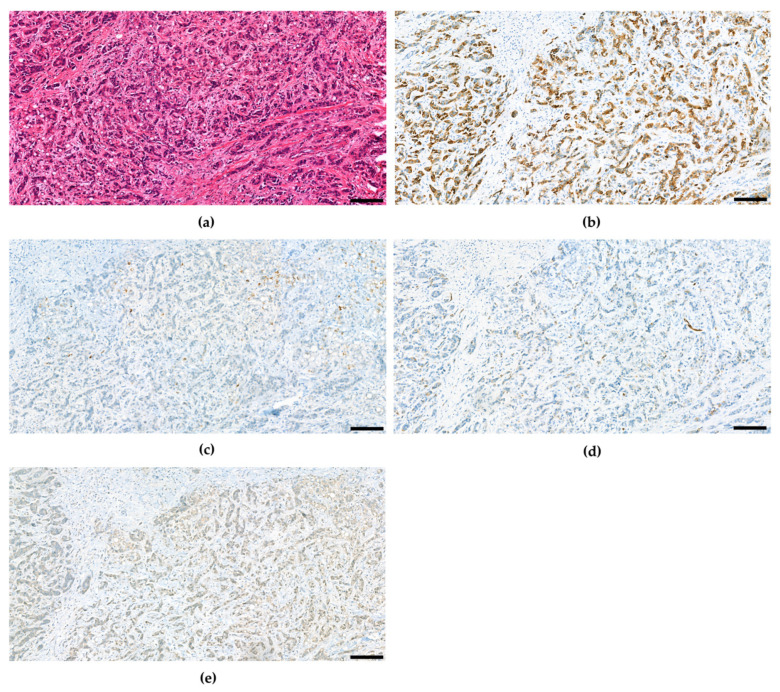
Combined hepatocellular and cholangiocellular carcinoma, intermediate cell component. CD56 and c-Kit were not expressed (not shown). (**a**) H&E. (**b**) CK7. (**c**) HepPar1. (**d**) CK19. (**e**) Arginase-1. Scale bar: 100 µm.

**Figure 5 cancers-15-00494-f005:**
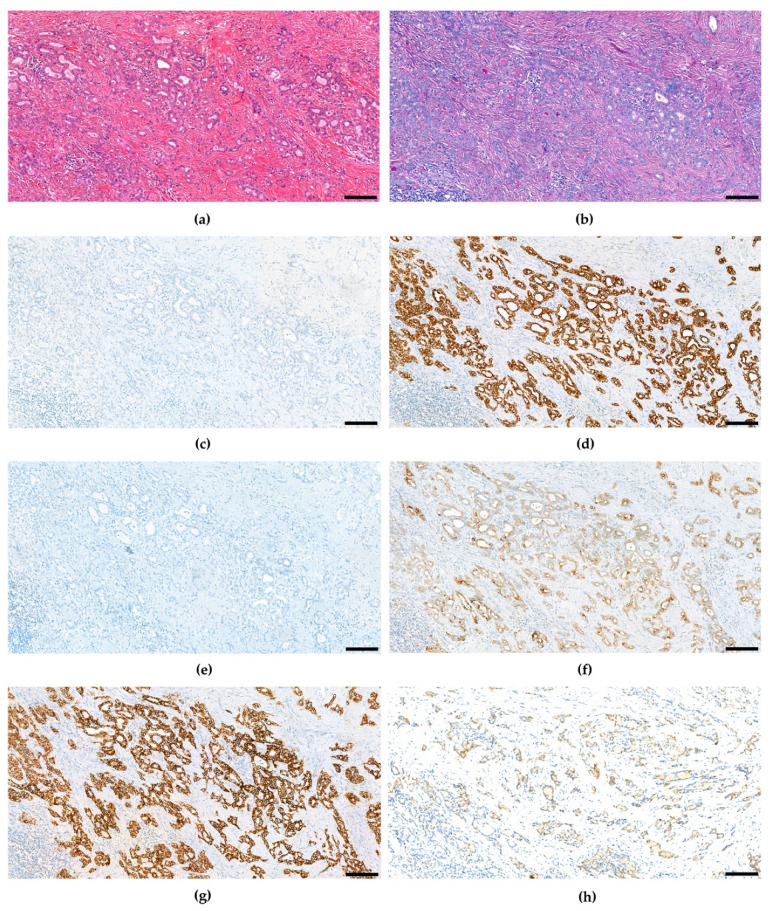
Combined hepatocellular and cholangiocellular carcinoma, cholangiocellular component (**a**) H&E. (**b**) PAS. (**c**) HepPar1. (**d**) CK7. (**e**) Arginase-1. (**f**) CK19. (**g**) EpCAM. (**h**) CD56. Scale bar: 100 µm.

**Figure 6 cancers-15-00494-f006:**
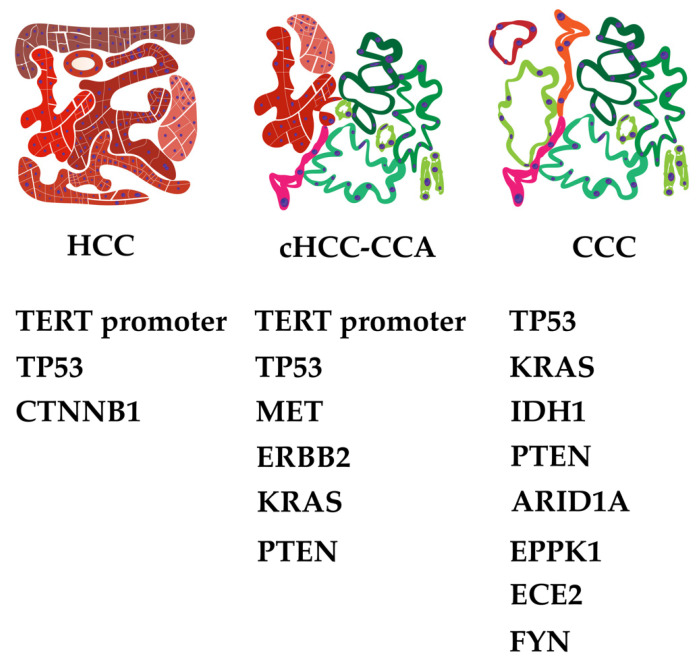
Most common mutational signatures in hepatocellular carcinoma (HCC), combined hepatocellular-cholangiocarcinoma (cHCC-CCA) and cholangiocarcinoma (CCC).

**Table 1 cancers-15-00494-t001:** Evolution of the World Health Organization (WHO) classification of combined HCC-CCA.

4th Edition (2010)	5th Edition (2019)
cHCC-CCA, classical type	cHCC-CCA, classical type
cHCC-CCC with stemcell-features, classical type	Intermediate cell carcinoma
cHCC-CCA with stemcell-features, intermediate typecHCC-CCA with stemcell-features, cholangiolocellular type	Included in the cholangiocarcinoma classification

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
