# Peer review of "Pathology of Combined Hepatocellular Carcinoma-Cholangiocarcinoma: An Update"

_cancers, 2023, doi:10.3390/cancers15020494_

Round 1

Reviewer 1 Report

* Introduction part should be enlarged based on the pathology of combined hepatocellular carcinoma-cholangiocar- 2 cinoma.

* It is also recommended to give different clinical classifications of the disease.

* It may be recommended to apply and report the scientific guideline in accordance with the research design.

Author Response

Reviewer #1:

1) Introduction part should be enlarged based on the pathology of combined hepatocellular carcinoma-cholangiocar- 2 cinoma.

2) It is also recommended to give different clinical classifications of the disease.

3) It may be recommended to apply and report the scientific guideline in accordance with the research design.

RESPONSE:

Ad 1) We agree with the reviewer that the introduction part should give a more comprehensive overview of combined hepatocellular and cholangiocellular carcinoma. We accordingly extended this part of the introduction.

Ad 2) Since combined hepatocellular and cholangiocellular carcinoma is classified after staging systems for cholangiocarcinoma and a few classification systems were developed and proposed over the course of time, we agree to discuss the Bismuth-Corlette system against the TNM staging system. We summarized this paragraph under a new built section “3.3 Basic clinical aspects”.

Ad 3) Because this is a narrative review approach on a rare entity, we state that by design we did not formally fulfill all PRISMA guidelines (e.g. data collection methods, synthesis methods). However, we give a transparent overview of how we conducted our literature work in the methods section. We also added the information that this is a narrative approach.

Reviewer 2 Report

The manuscript entitled “Pathology of combined hepatocellular carcinoma-cholangiocarcinoma: an update” by F. Roßner is a very well written review that summarizes the knowledge about some aspects of the combined hepatocellular carcinoma-cholangiocarcinoma, a rare tumor about which little is known. Author goes deeply about the pathology of this tumor, after providing some useful information about the tumor. The manuscript, as I said, is very well written, and it is very complete, with an extensive bibliography.

There are minor aspects that could be improved, and those are:

-I am not totally sure if the material and method section is necessary in a review. I would recommend suppressing it.

-please summarize the epidemiology findings.

-Some information about the progression of the disease would be interesting, years of survival after diagnosis, how aggressive the tumor is, does it metastasize? If it does it, which are preferential organs? Include also therapeutic options.

-organize the risk and prognostic factors, enumerate them separately.

-I would recommend eliminating table 1 as it can be confusing for the reader, as this classification is no longer used. I would suggest preparing a table in which current classification is depicted.

- A recapitulating/summary paragraph should be included at the end of 3.4.2 section.

- In table 3, it is stated that it has been modified after reference 131. Following the same concept as before, table should gather the most recent information. Also table 3 should include information about which of these markers are most accepted and which markers are being studied.

-in the discussion, the consensus reached by experts about the diagnosis of hepatocellular carcinoma-cholangiocarcinoma should be clearly stated.

-in the discussion, I do not quite understand why authors focus on the Hyppo pathway: it has not been mentioned before in the manuscript, and it is not the only pathway playing a role in this type of tumors.

Author Response

Reviewer #2:

The manuscript entitled “Pathology of combined hepatocellular carcinoma- cholangiocarcinoma: an update” by F. Roßner is a very well written review that summarizes the knowledge about some aspects of the combined hepatocellular carcinoma- cholangiocarcinoma, a rare tumor about which little is known. Author goes deeply about the pathology of this tumor, after providing some useful information about the tumor. The manuscript, as I said, is very well written, and it is very complete, with an extensive bibliography.

There are minor aspects that could be improved, and those are:

1) I am not totally sure if the material and method section is necessary in a review. I would recommend suppressing it.

2) Please summarize the epidemiology findings.

3) Some information about the progression of the disease would be interesting, years of survival after diagnosis, how aggressive the tumor is, does it metastasize? If it does it, which are preferential organs? Include also therapeutic options.

4) Organize the risk and prognostic factors, enumerate them separately.

5) I would recommend eliminating table 1 as it can be confusing for the reader, as this classification is no longer used. I would suggest preparing a table in which current classification is depicted.

6) A recapitulating/summary paragraph should be included at the end of 3.4.2 section.

7) In table 3, it is stated that it has been modified after reference 131. Following the same concept as before, table should gather the most recent information. Also table 3 should include information about which of these markers are most accepted and which markers are being studied.

8) In the discussion, the consensus reached by experts about the diagnosis of hepatocellular carcinoma-cholangiocarcinoma should be clearly stated.

9) In the discussion, I do not quite understand why authors focus on the Hippo pathway: it has not been mentioned before in the manuscript, and it is not the only pathway playing a role in this type of tumors.

RESPONSE:

Ad 1) Since this is a narrative approach for a review to objectively report the most current literature on combined hepatocellular and cholangiocellular carcinoma with emphasis on the pathology of this rare entity, we chose to add a methods section in order to give a transparent overview of how we conducted our literature work. We would therefore suggest to keep this section.

Ad 2) We totally agree with the reviewer about a short summary for better overview and added a conclusion at the end of paragraph 3.1

Ad 3) We agree with the reviewer about a short depiction of clinical aspects and created a new paragraph (3.3. Basic clinical aspects). In this paragraph, we now discuss presentation, progression, therapeutic options, prognosis, and staging in a brief manner, since this review rather focuses on pathology of the disease.

Ad 4) The paragraph 3.2 was revisited and revised. Risk factors and prognostic factors are now listed separately.

Ad 5) We agree with the reviewer that this table could be confusing to the reader since it is mainly based on knowledge of the 4th WHO edition. We therefore decided to eliminate the table without substitute. The evolution of the WHO classification is sufficiently discussed in paragraph 3.4.2 and an overview is depicted in Table 1.

Ad 6) Similar to point 2 of the review, we agree that a short summary of the most useful markers for cellular differentiation in diagnosis of cHCC-CCC would be helpful. We therefore now added a conclusive remark at the end of paragraph 3.5.2

Ad 7) We agree that a better summary of the most used biomarkers would be helpful, so we created a new section at the beginning of Table 2, which now depicts the biomarkers, which proved most useful in terms of specific expression patterns. Other studied biomarkers, which lack specificity are now separately depicted.

Ad 8) The discussion now summarizes the main consensus agreement of experts on the approach to a histopathological approach for combined hepatocellular and cholangiocellular carcinoma.

Ad 9) We agree that deeper discussion of underlying pathway alterations is not the primary focus of this work, so we eliminated the respective paragraph.

Round 2

Reviewer 1 Report

Thank you for this revision and changes were enough